# Electrochemical Performance of Chemically Activated Carbons from Sawdust as Supercapacitor Electrodes

**DOI:** 10.3390/nano12193391

**Published:** 2022-09-28

**Authors:** Meruyert Nazhipkyzy, Mukhtar Yeleuov, Shynggyskhan T. Sultakhan, Anar B. Maltay, Aizhan A. Zhaparova, Dana D. Assylkhanova, Renata R. Nemkayeva

**Affiliations:** 1Institute of Combustion Problems, Almaty 050012, Kazakhstan; 2Department of Chemical Physics and Material Science, Al-Farabi Kazakh National University, Almaty 050040, Kazakhstan; 3Department of Materials Science, Nanotechnology and Engineering Physics, Satbayev University, Almaty 050013, Kazakhstan; 4National Nanotechnology Laboratory of Open Type, Al-Farabi Kazakh National University, Almaty 050040, Kazakhstan

**Keywords:** sawdust, karagash, pine, activated carbon, supercapacitor

## Abstract

Activated carbons (ACs) have been the most widespread carbon materials used in supercapacitors (SCs) due to their easy processing methods, good electrical conductivity, and abundant porosity. For the manufacture of electrodes, the obtained activated carbon based on sawdust (karagash and pine) was mixed with conductive carbon and polyvinylidene fluoride as a binder, in ratios of 75% activated carbon, 10% conductive carbon black, and 15% polyvinylidene fluoride (PVDF) in an N-methyl pyrrolidinone solution, to form a slurry and applied to a titanium foil. The total mass of each electrode was limited to vary from 2.0 to 4.0 mg. After that, the electrodes fitted with the separator and electrolyte solution were symmetrically assembled into sandwich-type cell construction. The carbon’s electrochemical properties were evaluated using cyclic voltammetry (CV) and galvanostatic charge–discharge (CGD) studies in a two-electrode cell in 6M KOH. The CV and CGD measurements were realized at different scan rates (5–160 mV s^−1^) and current densities (0.1–2.0 A g^−1^) in the potential window of 1 V. ACs from KOH activation showed a high specific capacitance of 202 F g^−1^ for karagash sawdust and 161 F g^−1^ for pine sawdust at low mass loading of 1.15 mg cm^−2^ and scan rate of 5 mV s^−1^ in cyclic voltammetry test and 193 and 159 F g^−1^ at a gravimetric current density of 0.1 A g^−1^ in the galvanostatic charge–discharge test. The specific discharge capacitance is 177 and 131 F g^−1^ at a current density of 2 A g^−1^. Even at a relatively high scan rate of 160 mV s^−1^, a decent specific capacitance of 147 F g^−1^ and 114 F g^−1^ was obtained, leading to high energy densities of 26.0 and 22.1 W h kg^−1^ based on averaged electrode mass. Surface properties and the porous structure of the ACs were studied by scanning electron microscopy, energy-dispersive X-ray analysis, Raman spectroscopy, and the Brunauer–Emmett–Teller method.

## 1. Introduction

Renewable energy sources have been urged for continual improvement because of the increased demand for energy and depletion of fossil fuels. Therefore, the development of feasible energy storage devices is crucial to creating a more prosperous society. Due to their attractive advantages, including higher energy density and capacitance than conventional capacitors together with their high power density, and better life cycle than batteries, supercapacitors (SCs) are thought to be one of the most promising electrochemical energy storage solutions of the future [1,2,3,4,5,6,7,8,9,10]. To put it other words, SCs bridge the technology gap between batteries and dielectric capacitors due to their high power density along with high stability and reversibility [11,12]. It is known that the substantial capacitance produced by these systems results from a combination of the double-layer capacitance and pseudocapacitance associated with the participation of adsorbed intermediates in the surface redox-type processes [13]. The highly tunable properties of SCs and possibilities of coupling SCs with additional energy storage devices also serve as advantages for some applications such as hybrid and/or electric vehicles, wearable electronics, and stretchable devices [14,15,16].

It must be underlined that electrodes, current collectors, electrolyte, and spacers are the main components that should be brought up for maintaining efficient energy storage on SCs. Especially, electrodes that operate at the electrode/electrolyte interface are critically required to maximize the power density and cyclability of SCs [17]. Conducting polymers, metal oxides/nitrides, graphene, and several carbon materials are frequently employed as supercapacitor electrode materials. Carbon-based capacitors have drawn a lot of interest since they can have diversified morphologies and modification possibilities together with high stability and conductivity [18]. In the literature, graphene, carbon nanotube, and carbon nanofibers are reported as SC electrodes. It is known that graphene and carbon nanotube-based SCs offer great specific capacitance, excellent energy and power density, and can also increase the voltage window range, all of which enhance the material’s electrochemical performance; however, their low-cost mass production is still a great challenge [19,20,21]. Although the specific capacitance of the carbon fibers is very high, their unique structure makes it possible for them to be well-combined with conductive polymers such as polypyrrole, resulting in higher specific capacitance and energy density [22,23,24]. At this point, biomass-based porous carbons such as activated carbons seem attractive in terms of cost and sustainability [25]. Moreover, the amount of the charge stored, and its propagation during the charging/discharging, can differ significantly by altering the porous structure, size, and shape of pores and their functionalities [14].

With this perspective, the requirements for a cost-effective and environmentally benign approach from nonedible, abundant biomass sources have been anticipated to produce electrodes for SCs.

In [26], the electrode material was made from activated banana peels at a carbonization temperature of 900 °C using potassium hydroxide as an activating agent. It showed a specific capacity of 165 F g^−1^ at a current density of 0.5 A g^−1^ in a two-electrode cell.

The authors of work [27] carried out hydrothermal precarbonization and then pyrolysis of cucumber, transforming it into graphene-like carbon nanosheets. The obtained carbon nanosheets showed a specific capacity of 143 F g^−1^ at a current density of 0.2 A g^−1^ in an aqueous electrolyte 6 M KOH in the two-electrode cell setup.

In [28], biobased carbon materials from potato waste obtained by hydrothermal and thermochemical conversion technologies and used as electrode materials for supercapacitors showed an electrochemical performance of up to 134.15 F g^−1^.

Yakaboylu et al. [29] used the miscanthus grass biomass as the precursor for the sheet-like activated carbon synthesis and a specific capacitance up to 188 F g^−1^ at 0.1 A g^−1^ was achieved.

The waste wood-dust of Dalbergia sisoo (Sisau) was tested as a supercapacitor electrode base and showed a lesser specific capacitance of 104.4 F g^−1^ [30].

Activated potassium citrate obtained from 750 °C showed the best capacitance performance, exhibiting the highest specific capacitance of 242.5 F g^−1^ at 0.2 A g^−1^ [31].

Considering the performance of biomass-based carbonaceous electrodes has not reached an excellent level yet, determination of the ideal characteristics of each biomass material is essential to explore the appropriate methods for the activation technique as physical activation and chemical activation. Moreover, the carbonization pathway and parameters have a marked effect on the texture of the resultant activated carbon. For instance, hydrothermal carbonization (HTC) is a preferred technique to alter the morphology and porosity of biomass-based precursors under mild processing temperatures and self-generated pressure carbonization using pure water as a solvent [32,33,34]. Additionally, the combination of HTC with chemical activation may develop the desired specific surface area, porosity for specific applications, and enhance the yield of the carbons [35,36,37]. In this study, activated carbons were prepared by two different sawdust samples, karagash (type of elm which expanded in Kazakhstan) and pine, to compare the structural features and to determine the influence of the precursor and carbonization temperature on the capacitive performance of the porous carbon electrodes. Hydrothermal pretreatment and subsequent high-temperature activation were used to produce activated carbons and KOH served as a chemical activator in the process. To the best of our knowledge, no studies on the production of AC using combined HTC and KOH using these two regional biomass samples have been reported, specifically for electrode material in supercapacitor devices. Through the structural characterization as well as the chemical properties via electrochemical tests, the energy storage performance of the produced activated carbons as an electrode material was explored. The products were tested as electrodes in a two-electrode cell and the results from measurement of the impedance responses of the equivalent circuit model were obtained to explain physically significant parameters that showed the capacitive characteristics of the carbon electrodes.

This study showed that the use of electrode composites consisting of activated carbon derived from an available biomass precursor leads to improved performance of energy storage systems, in particular, electrochemical capacitors.

## 2. Experimental Procedure

### 2.1. Materials and Chemicals

Karagash sawdust was collected from Almaty (Kazakhstan). Then, it was crushed. The chemicals used in this study were of analytical grade: hydrochloric acid (HCl, 36.6%), potassium hydroxide (KOH, ≥85%, Sigma Aldrich, St. Louis, MO, USA), polyvinylidene fluoride (PVDF, EQ-Lib-PVDF, MTI Corporation), conductive carbon black (EQ-Lib-SuperC45, MTI Corporation), 1-methyl-2-pyrrolidone (NMP, ≥99.0%, Sigma Aldrich, St. Louis, MO, USA), argon (99.993% Ikhsan Technogas Ltd.), and titanium foil (MF-Ti-Foil-700L-105, MTI Corporation). Distilled water was used to prepare solutions and wash samples

### 2.2. Preparation of Activated Carbon

Activated carbons obtained from sawdust of karagash and pine were prepared by the following method. Sawdust (karagash, pine) was dried at 105 °C overnight. For the hydrothermal process, 5 g of sawdust was suspended in 100 mL of an aqueous solution of KOH (5 wt. %) in an autoclave and treated at 120 °C for 2 h. Then, the resulting suspension solution was separated by filtration through glass cloth (100 mesh). The resulting product was dried without washing at 105 °C to constant weight. This mass was crushed and carbonized at 700, 800, and 900 °C for 1 h in an inert argon gas at a heating rate of 5 °C min^−1^. The carbonized samples were washed with 2 M HCl solution and washed with distilled water until neutral acidity, and then dried at 100 °C to constant weight. The result was porous carbon from sawdust. Furthermore, the resulting material was ground by a mill in a pulsed mode. Next, the resulting material was separated by a 260 mesh sieve.

These obtained samples were designated as activated sawdust samples: AKS-700, AKS-800, AKS-900 and APS-700, APS-800, APS-900.

### 2.3. Characterization

Scanning electron microscopy (SEM, JEOL, model JSM-6490LA, FEI, USA) was used to determine the morphology of the samples. Energy-dispersive X-ray analysis (EDAX, JSM-6490LA, FEI, USA) was used for elemental analysis of the samples.

The specific surface area was characterized by the method of physical nitrogen adsorption at −196 °C with a surface area and adsorption analyzer (SORBTOMETR-M, Catakon, Novosibirsk, Russia). The surface area was calculated using the Brunauer–Emmett–Teller (BET) model.

### 2.4. Electrochemical Measurements

#### 2.4.1. Preparation of Electrodes

Activated carbon electrodes were prepared in the following way: activated carbon (75 wt.%) was mixed with a binder based on polyvinylidene fluoride (PVDF) (15 wt.%), carbon black (CB) (10 wt.%). |N-methyl-2- pyrrolidone (NMP) was used as a solvent for PVDF and was added into the mixture (AC:PVDF:CB) in an amount of 5.3 mL per gram of the mixture. The mixture was stirred for 20 min. The prepared suspension was applied to a titanium foil current collector on an area of 1 × 2 cm^2^. Then, the prepared electrodes were dried at 120 °C for 12 h.

#### 2.4.2. Electrochemical Measurements

To measure the electrochemical characteristics of the fabricated electrodes based on karagash and pine sawdust, a two-electrode cell with 6 M KOH was used as an electrolyte. The masses of the active material of the two electrodes were the same, and a symmetrical device was assembled from them to separate the two electrodes electrically. Cyclic voltammetry and galvanostatic charge–discharge measurements were carried out using an electrochemical workstation (galvanostat-potentiostat P-40X with an FRA-24M electrochemical impedance measurement module). The specific capacity was calculated from the curves of cyclic voltammetry and galvanostatic charge–discharge.

The CV data were obtained in a potential window of −0.1–0.9 V at scan rates of 5–160 mV s^−1^ in a 6 M KOH electrolyte. Equation (1) was applied to calculate the specific capacitance (*C_s_*) using cyclic voltammetry data:(1)$$\;C=\frac{A}{2\;m\;\nu \;\Delta V\;}$$
where *A* is the hysteresis area (*C*); m is the mass of the active substance of one electrode (g), that is, the mass of activated carbon or composite, which is 75% of the mixture; ν is the scan rate (*V* s^−1^); and Δ*V* is the potential window (*V*).

The specific capacity was also calculated from the GCD curves of the two-electrode cell using the equation:(2)$$C_{s}=\frac{2I\times t}{m\times \left(V_{2}-V_{1}\right)}$$
where *I* is the discharge current (*A*); t is the discharge time (*s*); *m* is the mass of the active substance of one electrode (g); and *V*_2_–*V*_1_ is the potential window (*V*).

For the impedance test, an alternating voltage with a constant amplitude of 10 mV was applied to the electrode with the frequency changing from 300 kHz to 10 MHz.

## 3. Results and Discussion

### 3.1. Activated Carbon Surface Characterization

Surface properties are important for materials for double-layer capacitors. The morphology of the APS-700, APS-800, APS-900 and AKS-700, AKS-800, AKS-900 was studied using SEM (Figure 1).

The irregular particle is clearly visible under SEM, as observed in Figure 2, and the images further reveal the well-developed porous structure of the samples. From SEM images, it is clearly seen that the pore sizes of AC from karagash sawdust were in the range of ~300 nm up to ~19 µm, while pore sizes of AC from pine sawdust were in the range of ~500 nm up to ~15 µm.

The pore development on the surfaces of the AC might be due to the course of dehydration of an activating agent, i.e., KOH. In addition to this, during the washing process, remaining alkali was removed, creating an empty space or pores. These structures could contribute to the ion transport by acting as charge storage interfaces.

Table 1 summarizes the specific surface area of the carbon samples which was calculated by BET method [38].

AC from pine sawdust has the highest specific surface area of 715 m^2^ g^−1^ compared to 557 m^2^ g^−1^ for the AC from karagash sawdust.

The elemental analysis of the samples was carried out by the EDAX method.

The elemental contents of the sawdust (karagash and pine) were evaluated by elemental analysis, as summarized in Table 2, indicating that the AC materials from the karagash sawdust are composed of C, O, K, Ca, Fe, and Ni, but the AC materials from the pine sawdust are composed of C, O, K.

The Raman spectra of sawdust based on karagash and pine are represented in Figure 2.

Degree of graphitization was calculated by the Formula (3) [39]
(3)$$G_{f}=\frac{A\left(G\right)}{\sum_{500}^{2000}A}\cdot 100\%$$
where $G_{f}$—degree of graphitization; *A*(*G*)—area of G peak; and ∑*A*—full area of the spectrum.

The dependence of the graphitization degree from temperature is shown in Figure 3.

The degree of graphitization calculated from the spectra increases with increasing carbonization temperature, and in the case of karagash, it reached a maximum value of ~25%. These carbonaceous materials with partial graphitization are highly suitable for application as electrodes due to their high electronic conductivities.

### 3.2. Electrochemical Characterization

The results of a comparative study of electrodes based on karagash and pine sawdust obtained at different activation temperatures (700, 800, 900 °C) are given in Table 3 and Table 4. It can be seen from the obtained data that the activation temperature is an important affecting parameter on the electrochemical characteristics of both AKS and APS-based electrodes. It has been stated that AKS-700 and APS-700 electrodes show the highest specific capacitance at various CV sweep rates (Table 3) and at various GCD current densities (Table 4) compared to activated carbons obtained at an activation temperature of 800 and 900 °C. This behavior can be mainly associated with a decrease in the capacitance of the electrical double layer due to a decrease in the available surface of the pore space for electrolyte ions with an increase in the activation temperature.

A general view of the experimental curves of cyclic voltammetry at different sweep rates (5–160 mV s^−1^) for electrode materials based on AKS-700 and APS-700 is shown in Figure 4A,B. The voltammetric characteristics of the AKS and APS electrodes were tested in the potential range of 0.0–1.0 V in a two-electrode cell configuration. The CV curves for both pairs of electrodes made on the basis of AKS-700 and APS-700 are almost rectangular in shape. This indicates that the materials act as electrical double-layer supercapacitors. It can be seen that the CV curve of the AKS-700 sample shows a larger surface area of the rectangle compared to the APS-700, which indicates the larger capacity of the AKS-700 sample (Figure 4).

Figure 5 shows the galvanostatic charge–discharge characteristics for electrode material based on AKS-700 and APS-700 in the potential range between 0 and 1 V in a 6 M KOH solution at different charge/discharge currents. The curves show approximately isosceles triangles, which are typical of electrical double-layer capacitance, not pseudocapacitance. This is also consistent with the information obtained from the CV curves (without a peak indicating Faraday processes) (Figure 4). It is clearly seen that the charge–discharge time for the AKS-700 electrodes is longer than for the APS-700 electrodes, indicating a higher specific capacity (Figure 5).

In Figure 5A, at a gravimetric current density of 100 mA g^−1^, the irreversible phenomenon, a long charge time with short discharge time, appeared. The irreversible phenomenon that the discharge time is shorter than the charging time is common in many supercapacitors at low current densities. It is very likely that the internal resistance of the device is relatively large, resulting in a significant voltage drop. In addition, the supercapacitor still has a severe self-discharge phenomenon, and even some other micro short circuits will lead to capacity loss and shorten the discharge time. This is one of the main drawbacks of the supercapacitor. The difference between charge and discharge times is negligible at high current densities. For this reason, supercapacitors are used in applications where high power density is required.

Figure 6A shows the specific capacitance of electrodes based on AKS-700 and APS-700 calculated from GCD curves at different discharge current densities. The specific capacitance of the obtained electrode materials is significantly higher at high current densities and scan rates even without additional modification of the materials [40,41,42]. The figure clearly shows that the specific capacitance of the AKS-700 electrode decreases slightly with increasing current density. This trend is characteristic of supercapacitors and reflects the fact that as the current density increases, some active surface areas become unsuitable for charge storage. However, in the studied electrodes, a slight decrease in capacitance with an increase in the charge–discharge current characterizes the best charge-exchange processes compared to traditional materials. Typically, pseudocapacitive materials show a higher capacitance loss as current density increases.

To study the fundamental behavior of the carbon in EDLCs, electrochemical impedance spectroscopy was performed by using an open-circuit potential. The semicircle at high frequency indicated the resistance from the diffusion/mass transfer of the ions through the porous carbon (Figure 7).

Solution resistance (Rs), which corresponds to the beginning of the semicircle, of the sample AKS-700 is 0.17 Ohm, whereas the Rs of the sample APS-700 is 0.23 Ohm. The magnitude of charge transfers resistance (Rct) can be obtained from the diameter of the semicircle. Rct of APS-700 is approximately 0.6 Ohm.

The electrochemical performance of activated carbon as the electrode active material of a supercapacitor depends on the porous structure, electrical conductivity, particle size distribution, etc. All these properties are mainly affected by the synthesis process and the nature of the precursors. Despite the fact that the synthesis of both precursors (AKS-700, APS-700) is the same, their initial properties are different. This is the reason why AKS and APS show different electrochemical characteristics. Further research should be aimed at understanding the influence of the initial properties of the starting materials on the characteristics of the final electrode materials.

## 4. Conclusions

Activated carbon from karagash (type of elm), which are expanding in Kazakhstan region, and pine sawdust was used as electrode for supercapacitors. Activated carbon from karagash as an electrode material for supercapacitors showed high rate capability and stable cyclability compared to the activated carbon from pine. Our experiment showed that karagash is a promising electrode material for supercapacitors.

This study provided a new approach to using the sawdust of karagash and pine trees to produce carbon material for EDLC application due to the low cost of biowaste material and simplicity of the activation process. The electrochemical property studies indicate the promising use of activated carbon from the sawdust of karagash and pine trees as an effective energy storage device, which curtails global energy demand and environmental concerns. At a relatively high scan rate of 160 mV s^−1^, a decent specific capacitance of 147 F g^−1^ and 114 F g^−1^ was obtained, leading to high energy densities of 26.0 and 22.1 W h kg^−1^ based on averaged electrode mass.

## Figures and Tables

**Figure 1 nanomaterials-12-03391-f001:**
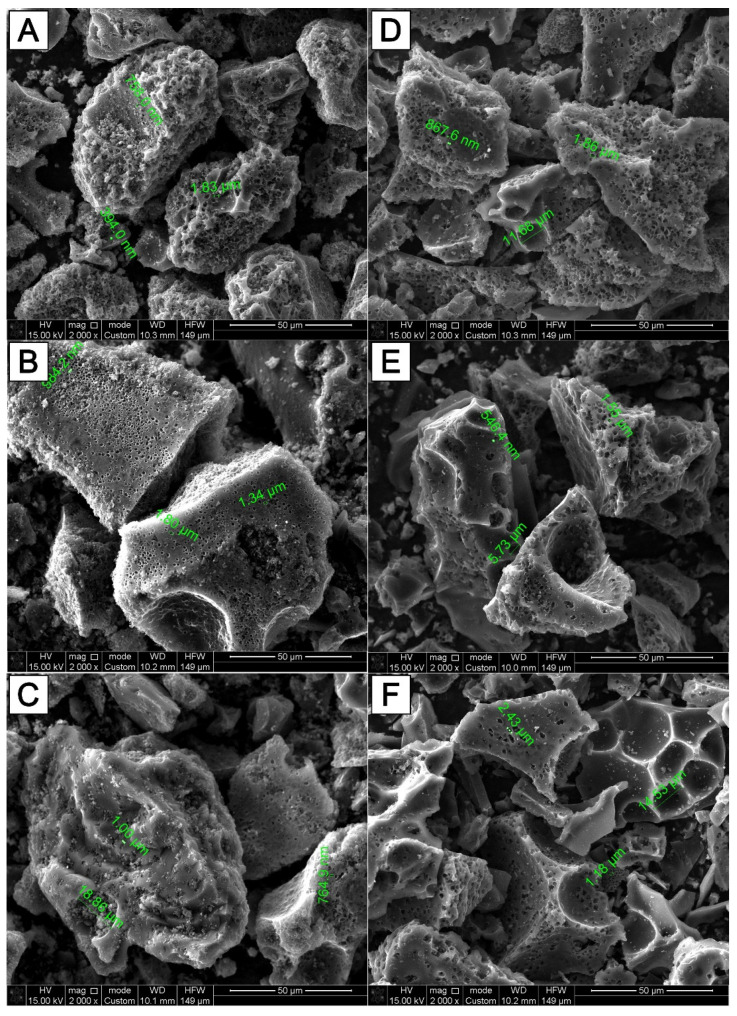
SEM images of carbon materials activated at different temperatures. (**A**) AKS-700 °C; (**B**) AKS-800 °C; (**C**) AKS-900 °C; (**D**) APS-700 °C; (**E**) APS-800 °C; (**F**) APS-900 °C.

**Figure 2 nanomaterials-12-03391-f002:**
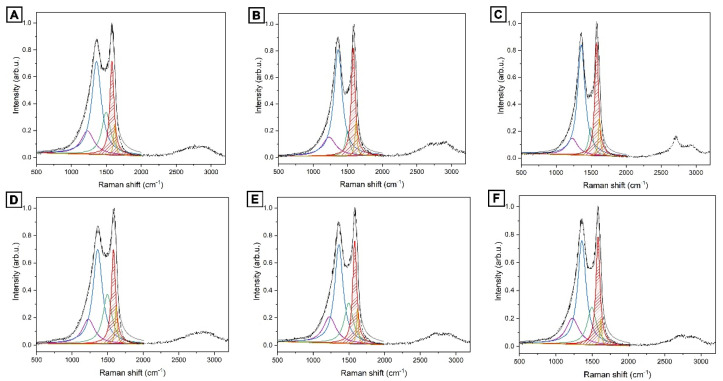
Raman spectra of a carbon material. (**A**) AKS-700 °C; (**B**) AKS-800 °C; (**C**) AKS-900 °C; (**D**) APS-700 °C; (**E**) APS-800 °C; (**F**) APS-900 °C.

**Figure 3 nanomaterials-12-03391-f003:**
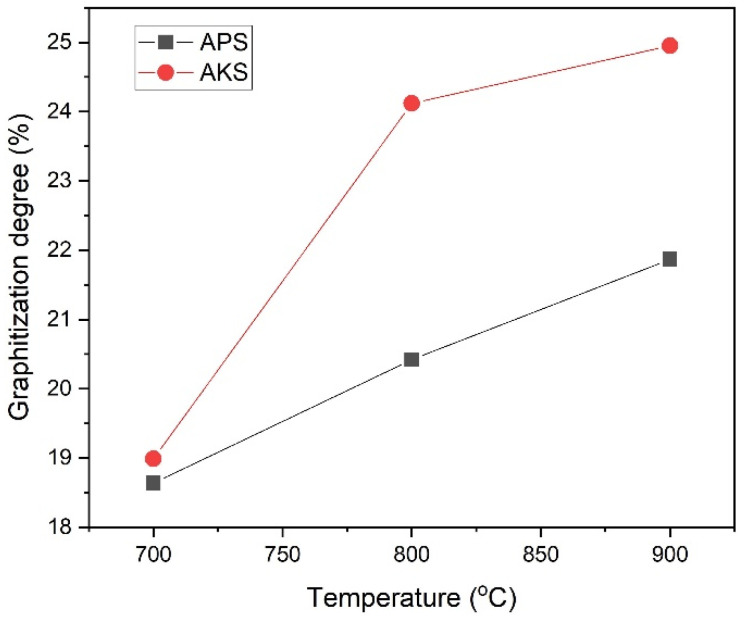
Dependence of graphitization degree from temperature.

**Figure 4 nanomaterials-12-03391-f004:**
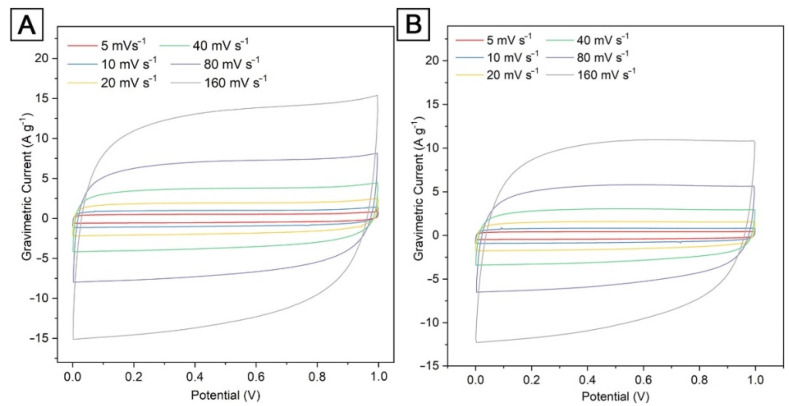
Cyclic voltammetry curves for electrodes based on (**A**) AKS-700, (**B**) APS-700, (**C**) AKS-800, (**D**) APS-800, (**E**) AKS-900 and (**F**) APS-900 at different scanning rates: 5, 10, 20, 40, 80, 160 mV s^−1^.

**Figure 5 nanomaterials-12-03391-f005:**
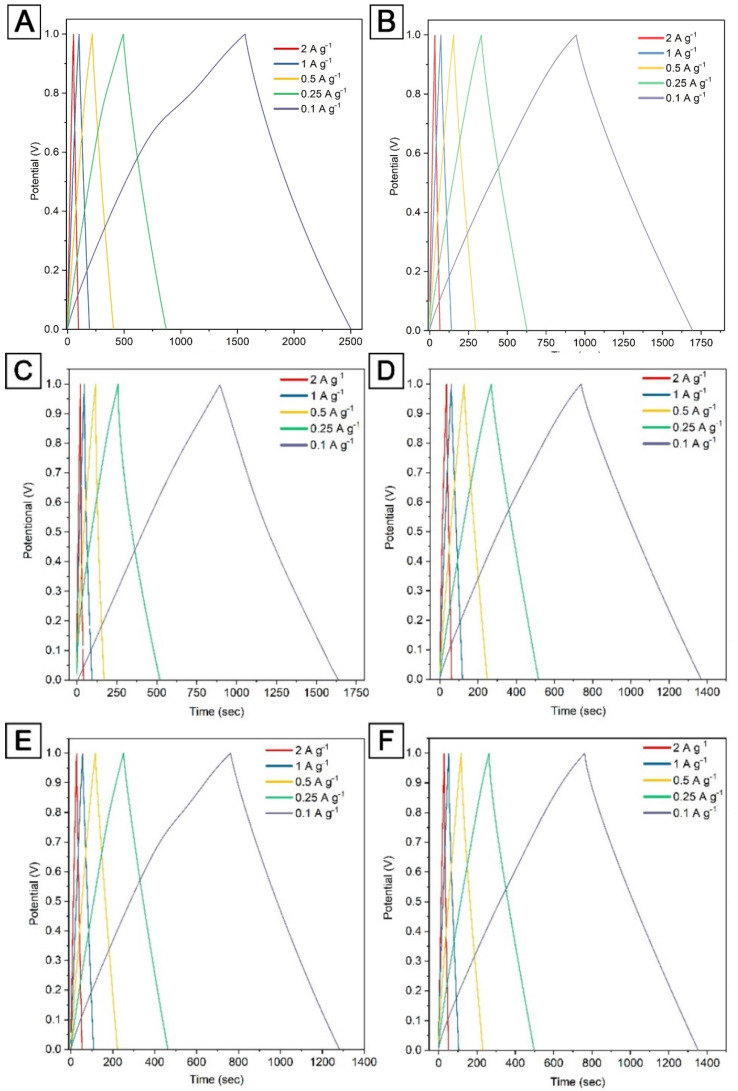
Curves of galvanostatic charge–discharge at different current densities (0.1, 0.25, 0.5, 1.0, 2.0 A g^−1^) for electrodes based on (**A**) AKS-700, (**B**) APS-700, (**C**) AKS-800, (**D**) APS-800, (**E**) AKS-900 and (**F**) APS-900 with 6 M KOH electrolyte.

**Figure 6 nanomaterials-12-03391-f006:**
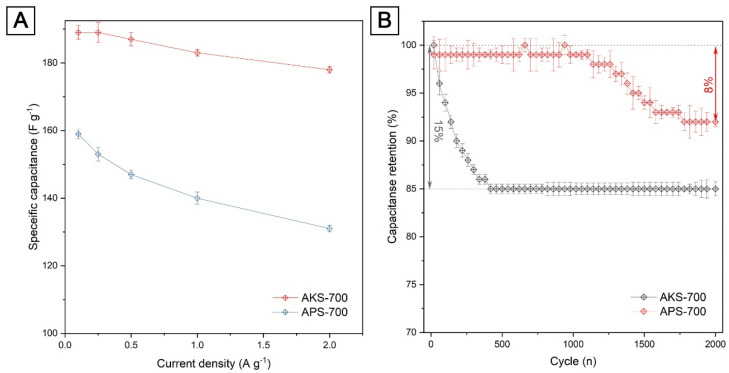
(**A**) The values of the specific capacitance of the electrodes measured at different current densities; (**B**) capacity retention of electrodes based on AKS-700 and APS-700 for 2000 cycles at constant current charge–discharge cycling, current density 1 A g^−1^.

**Figure 7 nanomaterials-12-03391-f007:**
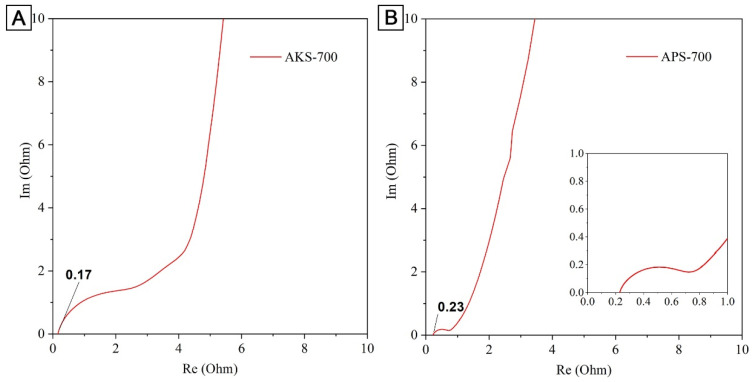
Nyquist plot for activated sawdust electrodes (**A**) AKS-700 and (**B**) APS-700.

**Table 1 nanomaterials-12-03391-t001:** The specific surface area of the activated carbons based on karagash and pine sawdust.

Sample	AKS-700	AKS-800	AKS-900	APS-700	APS-800	APS-900
S_BET_, (m^2^ g^−1^)	505	557	532	494	715	666

**Table 2 nanomaterials-12-03391-t002:** Elemental composition of samples AKS and APS (atomic concentration, at. %).

	C	O	K	Ca	Fe	Ni
**AKS-700**	90.29	7.30	1.80	0.57	−	−
**AKS-800**	85.37	14.63	−	−	1.80	0.57
**AKS-900**	84.69	10.90	1.17	−	1.41	1.83
**APS-700**	87.14	12.86	−	−	−	−
**APS-800**	93.87	6.13	−	−	−	−
**APS-900**	87.065	12.95	−	−	−	−

**Table 3 nanomaterials-12-03391-t003:** The specific capacitance of AKS and APS obtained from CV data at different scan rates.

Samples	Scan Rate, mV s^−1^					
	5	10	20	40	80	160
	Specific Capacitance, F g^−1^					
**AKS-700**	202	194	185	175	162	147
**AKS-800**	128	111	91	65	42	28
**AKS-900**	121	113	105	102	99	92
**APS-700**	161	155	148	139	127	114
**APS-800**	146	142	135	126	115	104
**APS-900**	115	108	99	87	73	59

**Table 4 nanomaterials-12-03391-t004:** The specific capacitance of AKS and APS calculated by the GCD data at different current densities.

Samples	Current Density, mA g^−1^				
	2000	1000	500	250	100
	Specific Capacitance, F g^−1^				
**AKS-700**	178	183	187	189	193
**AKS-800**	106	111	117	131	148
**AKS-900**	102	104	105	106	104
**APS-700**	131	140	147	153	159
**APS-800**	109	115	120	123	126
**APS-900**	91	104	112	118	118

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
