# Peer review of "Electrochemical Performance of Chemically Activated Carbons from Sawdust as Supercapacitor Electrodes"

_nanomaterials, 2022, doi:10.3390/nano12193391_

Round 1
Reviewer 1 Report
The work is good. However, several issues are needed to address as follows.
1. In Table 1, the author only displayed the specific areas, the pore size for the carbon samples at different temperatures should be provided. Besides, the font format should be aligned.
2. In fig. 5a, at a current density of 0.1 A g-1, the irreversible phenomenon, long charge time while short discharge time, appeared. The author should explain the detailed reason.
3. Most importantly, the author only reported the sample at 700 degrees, why not detect the samples of 800 and 900 degrees? I noticed that these samples had better specific areas and possess a tiny amount of Fe and Ni, which could provide a pseudo-capacitive contribution to the final performance.
4. References are not sufficiently cited. Several refs. on supercapacitors are suggested to cite, such as Small structures, 2022, 3, 2100115.
Author Response
The work is good. However, several issues are needed to address as follows.
- In Table 1, the author only displayed the specific areas, the pore size for the carbon samples at different temperatures should be provided. Besides, the font format should be aligned.
Author’s response:
We agree. But, unfortunately our BET equipment gives only surface area dates.
We had collaboration with Nazarbayev University. Their equipment was broken, unfortunately.
Therefore, in our manuscript we have used only specific surface area.
- In fig. 5a, at a current density of 0.1 A g-1, the irreversible phenomenon, long charge time while short discharge time, appeared. The author should explain the detailed reason.
Author’s response:
In figure 5a, at a gravimetric current density of 100 mA g-1, the irreversible phenomenon, long charge time while short discharge time, appeared. The irreversible phenomenon that the discharge time is shorter than the charging time is common in many supercapacitors at low current densities. It is very likely that the internal resistance of the device is relatively large, resulting in a significant voltage drop. In addition, the supercapacitor still has a severe self-discharge phenomenon, and even some other micro-short circuits will lead to capacity loss and shorten the discharge time. This is one of the main drawbacks of the supercapacitor. The difference between charge and discharge times is negligible at high current densities. For this reason, supercapacitors are used in applications where high power density is required.
- Most importantly, the author only reported the sample at 700 degrees, why not detect the samples of 800 and 900 degrees? I noticed that these samples had better specific areas and possess a tiny amount of Fe and Ni, which could provide a pseudo-capacitive contribution to the final performance.
Author’s response:
We have added results for 800 and 900 degree as Supplementary file.
- References are not sufficiently cited. Several refs. on supercapacitors are suggested to cite, such as Small structures, 2022, 3, 2100115.
Author’s response:
We agree. We have added additional references
Reviewer 2 Report
Authors synthesized carbon material using sawdust of karagash and pine trees, and proved that karagash is a promising electrode material for supercapacitors. In this study, the ACs presented high specific capacitance 202 F g-1 for karagash sawdust and 161 F g-1 for pine sawdust at constant current 0.1 A/g in the galvanostatic charge-discharge test. Even at a relatively high scan rate of 160 mV s-1, a high specific capacitance of 147 F g-1 and 114 F g-1 was obtained, leading to high energy densities of 26.0 and 22.1 W h kg-1 based on averaged electrode mass. This work offers a way for high performance capacitor material production, which is meaningful in this energy age. However, before it can be accepted for publication, the following comments should be solved.
1. Figure 2B is kind of confusion, please give an explanation why it’s so different with Figure 2A and 2C.
2. The image sequence annotations, for example a, b, c, are totally different in different figures. Please check and modify.
3. Figure resolution is poor, especially for figure 3, please change.
4. Please insert error bar into figure 6 to verify the reproducibility of the sample performances.
5. The electrochemical performances between AKS and APS are different, a comprehensive and reasonable explanation is necessary.
Author Response
- Figure 2B is kind of confusion, please give an explanation why it’s so different with Figure 2A and 2C.
Author’s response:
They were confusion. We have checked and changed
- The image sequence annotations, for example a, b, c, are totally different in different figures. Please check and modify.
Author’s response:
We have changed all figures on the same format
- Figure resolution is poor, especially for figure 3, please change.
Author’s response:
We have changed and sending original file of image
- Please insert error bar into figure 6 to verify the reproducibility of the sample performances.
Author’s response:
We have added error bar into Fig.6.
- The electrochemical performances between AKS and APS are different, a comprehensive and reasonable explanation is necessary.
Author’s response:
The electrochemical performance of activated carbon as the electrode active material of a supercapacitor depends on the porous structure, electrical conductivity, particle size distribution, etc. All these properties are mainly affected by the synthesis process, and nature of the precursors. Despite the fact that the synthesis of both precursors (AKS, AKS) is the same, their initial properties are different. We believe that this is the reason why ACS and APS show different electrochemical characteristics. Further research should be aimed at understanding the influence of the initial properties of the starting materials on the characteristics of the final electrode materials.
Reviewer 3 Report
In this manuscript, the authors synthesize activated carbon by carbonizing sawdust of karagash and pine with KOH activation. AKS-700 shows satisfactory electrochemical performance. At 0.1 A/g, the specific capacitance reached 193 F/g (two-electrode system). However, there are some issues in the present manuscript that need to be addressed. I suggest accepting this manuscript after major revisions by answering the questions below.
1. Biomass are promising carbon source for activated carbon materials. A lot of porous carbon materials have been developed from various biomass and applied in many fields. More discussions are suggested to be added in the introduction part and some related references are recommended to be cited: Journal of Bioresources and Bioproducts 2021, 6 (4), 292-322; Biochar 2022, 4, 17. https://doi.org/10.1007/s42773-022-00145-2. Biochar 2020, 2, 253–285.
2. The abstract part does not explain the influence of variables on the results, and lacks the expression of the conclusion.
3. The language need to be further polished. Expressions like “at low 1.15 mg cm2 mass loading” in line 22 should be revised.
4. The writing of units should be in the same style. For example, “0.1 A/g” and “1.15 mg cm2” should be revised to the same style.
5. The description of background in the introduction part lacks the support of research papers.
6. There is a sentence in Section 2.1 that is not related to this section, which is “At the same time, it is interesting that in the case of samples obtained on the basis of a Co catalyst, an explicit temperature dependence is not observed and their Raman spectra demonstrate relatively low values of FWHM(D) for almost all synthesis parameters – temperature and catalyst concentrations”.
7. In Table 1, the data are not arranged in a horizontal line.
8. More discussion about Fig. 2 are suggested to be added to reveal the influence of temperature on the porosity.
9. It is more professional to do N2 absorption/desorption isothermal to reveal the specific surface area and pore size distribution of activated carbon samples.
10. The figures in the main text start from Fig. 2, and Fig. 1 does not appear in the manuscript.
11. In Fig. 4, the degree of graphitization calculated from the spectra increases with increasing carbonization temperature. But from Table 3 and Table 4, I find that the capacitance performance decreases with the increase of carbonization temperature. Is there any connection between these two phenomena?
12. Please pay attention to different format in the charts. Such as “Roman shift, cm-1” and “Potential(V)”. Some of the diagrams have incorrect units, such as “5 mVs-1”, “10 Vs-1”, “0,5 A g-1”and so on. Figures 5, 6 and 7 have upper and right axes, but Figures 3 and 4 do not have these.
13. The date in the Table 3 and Table 4 can be impressed in the way of Fig. 6(a), which is more intuitive. Some date in the Table 3 and Table 4 are missing units.
14. It is suggested to do some comparisons on the capacitance performance of other biomass derived porous carbon materials. Some references are suggested to be cited: Biochar 2022, 4 (1), 50; ChemNanoMat 2021, 7 (1), 34-43.
15. The references are too less for a research paper. More papers published recently on capacitance performance of biomass derived carbon materials are suggested to be cited. For example, Diamond & Related Materials 2022, 128, 109247; Journal of Bioresources and Bioproducts 2021, 6 (2), 142-151.
Author Response
- Biomass are promising carbon source for activated carbon materials. A lot of porous carbon materials have been developed from various biomass and applied in many fields. More discussions are suggested to be added in the introduction part and some related references are recommended to be cited: Journal of Bioresources and Bioproducts 2021, 6 (4), 292-322; Biochar 2022, 4, 17. https://doi.org/10.1007/s42773-022-00145-2. Biochar 2020, 2, 253–285.
Author’s response:
We agree with reviewer. Introduction was expanded and references was added.
- The abstract part does not explain the influence of variables on the results, and lacks the expression of the conclusion.
Author’s response:
Abstract was extended.
- The language need to be further polished. Expressions like “at low mass loading 1.15 mg cm2” in line 22 should be revised.
Author’s response:
English and grammar were checked and revised throughout the article. Expressions “at low 1.15 mg cm2 mass loading” in the abstract has been revised and modified.
- The writing of units should be in the same style. For example, “0.1 A/g” and “1.15 mg cm2” should be revised to the same style.
Author’s response:
Corrections has been made accordingly. All units have been reviewed throughout the article and changed. Now all units are in the same style (e.g. A g-1, F g-1, mV s-1, m2 g-1)
- The description of background in the introduction part lacks the support of research papers.
Author’s response:
We agree. It was considered research papers in the introduction part.
- There is a sentence in Section 2.1 that is not related to this section, which is “At the same time, it is interesting that in the case of samples obtained on the basis of a Co catalyst, an explicit temperature dependence is not observed and their Raman spectra demonstrate relatively low values of FWHM(D) for almost all synthesis parameters – temperature and catalyst concentrations”.
Author’s response:
We agree. There were mistake. That sentence was removed.
- In Table 1, the data are not arranged in a horizontal line.
Author’s response:
We agree. It was corrected.
- More discussion about Fig. 2 are suggested to be added to reveal the influence of temperature on the porosity.
Author’s response:
It was added discussion according to Fig.2.
- It is more professional to do N2 absorption/desorption isothermal to reveal the specific surface area and pore size distribution of activated carbon samples.
Author’s response:
We agree. But, unfortunately our BET equipment gives only surface area dates.
We had collaboration with Nazarbayev University. Their equipment was broken, unfortunately.
Therefore, in our manuscript we used only surface area.
- The figures in the main text start from Fig. 2, and Fig. 1 does not appear in the manuscript.
Author’s response:
Yes. We have corrected.
- In Fig. 4, the degree of graphitization calculated from the spectra increases with increasing carbonization temperature. But from Table 3 and Table 4, I find that the capacitance performance decreases with the increase of carbonization temperature. Is there any connection between these two phenomena?
Author’s response:
According to the Raman spectra, the degree of graphitization increases with increasing carbonization temperature while the specific capacitance of both materials (AKS, AKS) decreases with increasing carbonization temperature. Graphitization should increase the electrochemical capacitance because it increases the conductivity of the carbon material. However, in our study, a different phenomenon is observed. Changes in the porous structure of carbon materials can be the main reason for the decrease in specific capacity with increasing carbonization temperature.
- Please pay attention to different format in the charts. Such as “Roman shift, cm-1” and “Potential(V)”. Some of the diagrams have incorrect units, such as “5 mVs-1”, “10 Vs-1”, “0,5 A g-1”and so on. Figures 5, 6 and 7 have upper and right axes, but Figures 3 and 4 do not have these.
Author’s response:
We have corrected to the same format units and added right axes for Figures 3 and 4.
- The date in the Table 3 and Table 4 can be impressed in the way of Fig. 6(a), which is more intuitive. Some date in the Table 3 and Table 4 are missing units.
Author’s response:
We appreciate the reviewer’s suggestions about the table 3 and 4. The missing units were put into tables.
- It is suggested to do some comparisons on the capacitance performance of other biomass derived porous carbon materials. Some references are suggested to be cited: Biochar 2022, 4 (1), 50; ChemNanoMat 2021, 7 (1), 34-43.
Author’s response:
In order to show the significance of the proposed electrode materials, several works by other authors have been added for comparative purposes. [36. Xia, Chengkai, et al. "A sulfur self‐doped multifunctional biochar catalyst for overall water splitting and a supercapacitor from Camellia japonica flowers." Carbon Energy (2022).; 37. Ahmed, Sultan, Ahsan Ahmed, and M. Rafat. "Investigation on activated carbon derived from biomass Butnea monosperma and its application as a high performance supercapacitor electrode." Journal of Energy Storage 26 (2019): 100988.; 38. Cao, L., Li, H., Xu, Z., Gao, R., Wang, S., Zhang, G., ... & Hou, H. (2021). Camellia pollen‐derived carbon with controllable N content for high‐performance supercapacitors by ammonium chloride activation and dual N‐doping. ChemNanoMat, 7
- The references are too less for a research paper. More papers published recently on capacitance performance of biomass derived carbon materials are suggested to be cited. For example, Diamond & Related Materials 2022, 128, 109247; Journal of Bioresources and Bioproducts 2021, 6 (2), 142-151.
Author’s response:
We have expanded references till 39.

Round 2
Reviewer 3 Report
The manuscript has been revised according to the comments, but there are some important issues unsolved. Major revisions are required.
1. More references are suggested to be added in the introduction for broad readers, for example Journal of Bioresources and Bioproducts 2021, 6 (2), 142-151; Diamond & Related Materials 2022, 128, 109247; Chemical Engineering Journal, 2023, 451, 138877.
2. Nyquist plot in Figure 7 need to be revised. The X axial and Y axial should show the same scale.
3. There are too many labels on X axial and Y axial in Figure 4a, 5a, 5b, 6a, 7a and 7b. Please redraw those figures.
4. Please pay attention to the writing of subscripts and superscripts. For example, "mA g-1" in line 272 need to be revised.
5. Why the charge/discharge current density for AKS-700 and APS-700 is limited to 2 A g-1? Did the authors tried higher current density?
6. The stability for AKS-700 and APS-700 is poor. Usually, more than 10000 cycles is required for carbon electrodes and the decrese is less than 10%. Please give some explainations why stability for AKS-700 and APS-700 is poor.
7. page numbers for reference 7 are missing. Please double check all the references.
Author Response
The manuscript has been revised according to the comments, but there are some important issues unsolved. Major revisions are required.
- More references are suggested to be added in the introduction for broad readers, for example Journal of Bioresources and Bioproducts 2021, 6 (2), 142-151; Diamond & Related Materials 2022, 128, 109247; Chemical Engineering Journal, 2023, 451, 138877.
Author’s response: We agree and we have added suggested publications.
- Nyquist plot in Figure 7 need to be revised. The X axial and Y axial should show the same scale.
Author’s response: We have revised Figure 7.
- There are too many labels on X axial and Y axial in Figure 4a, 5a, 5b, 6a, 7a and 7b. Please redraw those figures.
Author’s response: We have redrawn next Figures 4a, 5a, 5b, 6a, 7a and 7b.
- Please pay attention to the writing of subscripts and superscripts. For example, "mA g-1" in line 272 need to be revised.
Author’s response: We agree.
- Why the charge/discharge current density for AKS-700 and APS-700 is limited to 2 A g-1? Did the authors tried higher current density?
Author’s response: We only performed a galvanostatic charge-discharge test at higher current densities (5 A/g and 10 A/g) for the sample AKS-700. We didn't include it because we don't have test results for the sample APS-700 at such high current densities.
- The stability for AKS-700 and APS-700 is poor. Usually, more than 10000 cycles is required for carbon electrodes and the decrese is less than 10%. Please give some explainations why stability for AKS-700 and APS-700 is poor.
Author’s response: We think the stability issue is related to the experimental setup. We used a simple two-electrode setup in which the electrolyte was open in the air, which caused the water-based electrolyte to dry out partially during the long charge-discharge cycle test. Unfortunately, we don't have a coin cell assembly line in our lab. The electrochemical test set-up is given below.
- page numbers for reference 7 are missing. Please double check all the references.
Author’s response: We have added page number.

Round 3
Reviewer 3 Report
The manuscript is revised according to the comments and could be accepted now.